# Table Tennis as a Tool for Physical Education and Health Promotion in Primary Schools: A Systematic Review

**DOI:** 10.3390/sports13080251

**Published:** 2025-07-31

**Authors:** M. A. Ortega-Zayas, A. J. Cardona-Linares, M. Lecina, N. Ochiana, A. García-Giménez, F. Pradas

**Affiliations:** 1ENFYRED Research Group, University of Zaragoza, 22001 Huesca, Spain; maortega@unizar.es (M.A.O.-Z.); acardona@unizar.es (A.J.C.-L.); alejandro.garcia@unizar.es (A.G.-G.); franprad@unizar.es (F.P.); 2Department of Physical Education and Sports Performance, Faculty of Movement, Sports and Health Sciences, Vasile Alecsandri University of Bacău, 600115 Bacău, Romania; sochiana@ub.ro

**Keywords:** health promotion, teaching strategies, children’s physical activity, motor skills, school-based interventions

## Abstract

Racket sports are very popular not only in professional sports but also among recreational players. As a result, their impact on the field of education has grown significantly in recent years. Table tennis (TT) offers significant advantages in terms of skill acquisition and health improvement. Nevertheless, its application within physical education (PE) curricula remains undetermined. The aim of this systematic review was to analyze the use of TT as educational content in the subject of PE in primary school. The PRISMA 2020 methodology was used to conduct the systematic review. Six databases (ERIC, Pubmed, ScienceDirect, Scopus, Sport Discus, and Web of Science) were used during the search process. The search cutoff date was December 31, 2024. After applying the eligibility criteria, 3595 articles were found. Only seven studies were selected for the final analysis and the data included 1526 students from primary school. The results indicate that TT is rarely used in primary education during PE classes. Research indicates interest among teachers and students in playing it during PE. Furthermore, due to the benefits, motivation, and interest this sport generates, educational experiences have been developed, such as roundtable discussions, table top tennis, balloon TT, and TT triathlon. A lack of materials, facilities, and teacher training for teaching this sport is notable. The teaching experiences analyzed in this review confirm that TT is a highly versatile and interesting sport as a subject matter in school PE. The use of TT allows for innovative, comprehensive, and inclusive PE, thanks to the sport’s multiple possibilities. Given its adaptability and low entry barrier, TT could serve as an effective tool for increasing children’s physical activity levels, improving motor skills, and fostering social–emotional development. However further research is needed to quantify its impact on health outcomes such as cardiovascular fitness or obesity prevention.

## 1. Introduction

In recent decades, the education trend, particularly in physical education (PE), has shifted towards providing students with a wide range of physical activities and motor experiences, as engaging in meaningful PE promotes the inclusion of new activities, which in turn increases students’ motivation [1]. The goal is to enhance their understanding and promote comprehensive motor, intellectual, and emotional development [2,3]. This approach aims to contribute to personal growth and improve the quality of life by supporting students in developing their capacities and skills, fostering continuous improvement. It involves expanding their movement possibilities and deepening their understanding of their motor skills, which encourages positive attitudes, values, and norms related to physical activity and motor behavior [4,5].

Table tennis (TT) is a highly popular sport, recreationally and competitively, and was accepted as an Olympic discipline by the International Olympic Committee in 1988. In educational settings, TT has gained recognition as a valuable teaching tool over the years for its multiple benefits in the school environment, being recognized as a sport for everyone and for increasing quality of life, easy to learn at a basic level, versatile because it can be played almost anywhere, easy to adapt if necessary by allowing modified tables to be used for its game, which does not depend on the weather, relatively economical, and a healthy sport, especially of interest to develop concentration and coordination [6,7,8].

The effectiveness of PE in improving students’ physical condition and motor skills [9] has been historically demonstrated in multiple studies and is part of the justification for its permanence in official curricula [10]. Different authors point out that TT can be effective educational content, highlighting its ability to improve both motor skills and intellectual and affective development [11,12] even in players with some disabilities including developmental coordination disorder [13] or in adolescents with attention-deficit/hyperactivity disorder (ADHD) [7,8,9,14,15]. In addition, it has been shown that the practice of TT offers interesting opportunities to maintain and improve socio-emotional well-being [14,16], produces different benefits on motor development from an early age [17], and develops and perfects physical capabilities [18] and health of the general population [19,20,21].

But TT also offers improvements in academic achievement [22]: numerous studies have explored the multi-dimensional benefits of TT, highlighting its potential to foster advancements in various aspects of learning, with this sport recognized for its potential to improve different aspects of learning. In this sense, authors such as Wulf et al. [23] have shown that using TT in educational contexts can enhance attention and memory and, as Wu et al. [24] points out, promote self-esteem and group cohesion among young people, favoring the development of attention and memory [25], besides improving motor skills and academic performance [22].

On the other hand, some authors point out that TT, in addition to developing motor skills, also produces a significant improvement in executive function and social behavior in both children and adolescents [26]. Recent studies, such as those by Elferink et al. [27], have provided a deeper insight into the pedagogical effects of this sport in the educational field, evidencing its positive impact on students’ perceived competence and learning. Finally, Aljaafreh [28] in a study carried out on schoolchildren shows that TT improves the speed and precision of visual reactions and the volume of information, observing a positive impact on coordination, even in adolescents with Down syndrome [8]. These studies suggest that TT is not only effective in improving motor skills but also plays a crucial role in the development of social and cognitive skills [11,12,13,15,21,23,24]. As can be seen, the practice of TT seems to have positive effects on different aspects related to learning, motor development, and health [9,21]. However, despite the benefits of the practice of this sport, at an educational level, the impact that TT can have as pedagogical content in PE classes is unknown. In this sense, more precise research is needed to delve into the levels of use of this sport in the initial educational stages in this area of knowledge. This systematic review aims to find out if table tennis is used in schools, particularly during PE classes, and analyze why it is selected.

## 2. Materials and Methods

A systematic search was conducted following the PRISMA method [29] and was registered with the International Prospective Register of Systematic Reviews (PROSPERO; CRD42024545478). The systematic review was written following the Preferred Reported Items for Systematic Review and Meta-Analysis Protocols (PRISMA-P) guidelines [30].

### 2.1. Eligibility Criteria for Study Selection

The PICO (Population, Intervention, Comparison and Outcome) framework was used to set the objective of this systematic review according to PICO guidelines [31,32]. The eligibility criteria are fully described in Table 1.

### 2.2. Search Strategy

Six databases were scrutinized, with a deadline of 31 December 2024. The equations used in every database are as follows:ERIC (“table tennis,” OR “ping-pong,”) AND “physical education” OR “primary” OR “secondary” OR “university”.PubMed = (((“table tennis” OR “pigpong”) AND (“physical education”)) AND (“primary” OR “secondary” OR “university”)) Filters: Adaptive Clinical Trial, Case Reports, Classical Article, Clinical Study, Clinical Trial, Clinical Trial Protocol, Clinical Trial, Phase I, Clinical Trial, Phase II, Clinical Trial, Phase III, Clinical Trial, Phase IV, Controlled Clinical Trial, Multicenter Study, Observational Study, Randomized Controlled Trial, Twin Study, Validation Study, English, Spanish, Humans, Female, Male, Child: birth–18 years, Child: 6–12 years, Adolescent: 13–18 years, Adult: 19+ years, Young Adult: 19–24 years, Exclude preprints.ScienceDirect ((“table tennis,” OR “ping-pong,”) AND (“physical education”)).SCOPUS TITLE-ABS-KEY (((“table tennis” OR “ping-pong”) AND (physical AND education) AND (“primary” OR “secondary” OR “university”) AND (LIMIT-TO (SRCTYPE, “j”)) AND (LIMIT-TO (OA, “all”)) AND (LIMIT-TO (PUBSTAGE, “final”)) AND (LIMIT-TO (DOCTYPE, “ar”)) AND (LIMIT-TO (LANGUAGE, “English”) OR LIMIT-TO (LANGUAGE, “Spanish”)) AND (LIMIT-TO (EXACTKEYWORD, “Table Tennis”)).SPORTDiscus ALL = ((“table tennis,” OR “ping-pong”) AND (“physical education”))Web of table tennis OR ping-pong (Title) and physical education (All Fields) and primary or secondary or university (All Fields) and Article (Document Types) and English or Spanish (Languages) and Article (Document Types) and All Open Access (Open Access).

### 2.3. Data Extraction

All articles sought for retrieval (*n* = 230) were downloaded and their data was extracted into an Excel spreadsheet (Microsoft Corporation, Redmond, WA, USA) for the review. During this process, the titles and abstracts of the articles that met the eligibility criteria were evaluated according to the established inclusion/exclusion criteria (Table 1).

### 2.4. Risk of Bias

Two independent reviewers (F.P.) and (M.O.) selected the included papers based on established criteria. Mendeley Desktop^®^ v.2112.0 (Elsevier, Amsterdam, The Netherlands) was used to remove duplicate articles and analyze titles and abstracts. When necessary, a further full-text analysis was conducted. Both reviewers always approved the decisions. However, a third reviewer was consulted to solve disagreements. The research and analysis process lasted a total of three weeks.

### 2.5. Quality Assessment of Studies Included

The type of studies included in this systematic review (i.e., non-intervention, observational, non-controlled trials, and descriptive) along, with the focus on the educational field, recommend the use of a specific tool designed and validated for the analysis of the quality of the studies included in this systematic review. Most common tools, such as Cochrane [33] or the Physiotherapy Evidence Database scale (PEDro) [29], are not suitable for these types of studies. We applied the critical appraisal checklist for an article on an education intervention from the University of Glasgow [34] adapted from Morrison et al. [35]. This tool includes 13 questions, classified into four sections (q1, the question of the study; q2–q9, the validity of the results; q10–q11, precision in the results; and finally, q13–q14, the applicability of the study in a real educational setting). For a detailed list of all the items, see Appendix A. Every question must be answered with yes, cannot tell, or no (Figure 1). The tool does not use the number of “yes” answers to establish each of the categories but leaves the decision to define cutoff values for each category up to the user. The authors agreed on classifying the seven articles as good when 11 or more items were marked “yes”, fair when it was 7 to 10 items, and poor when only 6 or fewer items were marked “yes”.

## 3. Results

### 3.1. Main Search

A total of 3595 original articles were initially selected, with 83 of them being duplicates. Once the duplicate items were removed, a total of 3512 items were identified. Once the titles and abstracts were reviewed, 3282 articles were excluded because they did not meet any of the five established inclusion criteria. The full texts of the remaining 230 articles were analyzed. In the next screening phase, 201 articles were excluded because they did not meet inclusion criteria numbers 1 and 2, and a total of 29 articles were evaluated for admissibility, of which 22 were eliminated in the double reading process. In this way, a total of 7 articles met all the eligibility criteria and were included in the final qualitative synthesis (Figure 2).

Once the studies that met the objective and inclusion criteria proposed for this research were selected, the most relevant information was extracted from those variables considered of interest for this systematic review (Table 1).

### 3.2. Results of Studies

The seven articles included in the systematic review are described in Table 2.

## 4. Discussion

This systematic review aims to find out if TT is used in schools, particularly during PE classes, and analyze why it is selected. Seven studies were finally included: three studies were descriptive [36,37,38] and four were didactic experiences [39,40,41,42] related to the use of TT, or modified and adapted games related to this sport in the field of PE in the primary education stage. To our knowledge, this systematic review is the only that summarizes the evidence of the use of TT in primary schools, along with its limitations and successful practical experiences.

The research conducted by Greenwood et al. (2001) [36] is one of three descriptive studies found in this review. This research raises the question of how the selection of educational content should be to meet the general objectives of the curriculum, focusing on knowing the preferences of 751 students about the sports content used in PE classes. The analysis of the results obtained reveals that in both genders, 32.4% of students value the sport of TT of great interest as sports content in PE classes, being valued more positively by the male sample (33.8%) than by the female sample (30.9%), in a similar way to other studies [43,44].

The results obtained by Greenwood et al. (2001) [36] indicate that it would seem logical to think that PE teachers should consider TT as one of the educational contents to be incorporated into their classes, as this sport arouses interest among students, even those with some type of physical disability [45], allowing physical activity to be promoted [44]. One of the most interesting reflections of this research points towards the idea that if students are ignored in the curriculum development process, PE teachers might not be able to provide the experiences that best meet the movement needs of their students [46]. With this argument in mind, Greenwood et al. (2001) [36] point out that it would be reasonable to allow students to have a say in which activities should be included in the PE curriculum in order for it to be truly educational. From this perspective, the immediate quality of the movement experiences provided to students should be a criterion for the selection of sports activities within the PE curriculum. In this sense, incorporating TT into the school curriculum could improve students’ motivation, health, and motor skills, aligning with educational objectives, as indicated by other studies [44].

However, the authors warn that the PE teacher, in addition to considering the interests of the students, must also be cautious when selecting their content so that it can be taught within the specific limitations of the school system, considering, for example, the facilities and space available, as well as the interest of that sports activity in a given geographical area. In short, according to Greenwood et al. (2001) [36], the PE professional must strike a balance between the wishes of the students and the feasibility in practice. This dilemma invites reflection on the extent to which PE teachers should prioritize students’ preferences, such as incorporating the sport of TT as educational content, over traditional pedagogical objectives, suggesting a hybrid model that combines curricular physical sports activities with other types of skills to ensure comprehensive motor development [17], without sacrificing the enthusiasm of schoolchildren. The study by Hoffmann et al. (2018) [37] analyzes the teaching of racket sports (RS) in a sample of 551 teachers of both sexes (male = 97; female = 401), with a contrasted experience (17.00 ± 10.86 years), who teach PE in primary education in Germany. This study is of great interest as it is perfectly complemented by the one carried out by Greenwood et al. (2001) [36], where the students’ vision is given, allowing us to know both positions, students and teachers of PE in the field of primary education.

The findings obtained in this research show that 67.97% of the schools and 69.88% of the teachers teach different RS (badminton, tennis, TT, and basic games with rackets). The results of the study by Hoffmann et al. (2018) [37] indicate that TT is used as educational content in PE classes by 20.69% of teachers, ranking third among the most used RS, behind badminton (54.6%) and basic racket games (59.77%) but ahead of tennis (14.66%). The results of this study point towards the idea that TT in Germany is of interest to be used as content in PE classes, a fact that is confirmed by verifying with quantitative data that its choice as educational content by teachers has increased in the last decade, since a few years ago its frequency of use was lower and stood at 13.3% [47]. One of the most interesting findings of this research is that almost 70% of the primary school teachers participating in this study use one or more RS as content in their PE classes. However, the authors highlight that those aspects, such as being a specialist in PE, practicing RS in free time, work experience, the school’s internal PE curriculum, and RS in the school’s internal PE curriculum, show a significant impact on the selection and teaching of RS as content (Herrero et al., 2016) [48]. This study shows that the gender variable of teachers has no relationship with the fact of teaching RS in PE classes, as indicated by similar studies [49].

On the other hand, this research confirms that being a specialist teacher in PE has a greater impact on the use of different RS in PE classes. This result is of great interest, as it indicates a greater knowledge of PE specialists on specific motor skills, an issue that allows for improving the quality of teaching for young people [50]. Work experience is another fundamental factor that significantly influences the implementation of the content of different RS, in general, and TT in particular in PE classes [48,51]. Finally, the authors of this study highlight that it seems reasonable to think that teachers who have a personal interest or preference for the practice of an RS in their free time, such as TT, more frequently transfer the practice of this sport as teaching content to their PE classes [48,49]. Xiao et al. (2020) [38], in their research, explore the factors that affect TT skills at school age, with the intention of establishing the levels of intrinsic motivation and the persistence of the young population in the performance of future physical activity. The study was carried out on 1526 students (age = 12.31 ± 1.32 years) belonging to 755 primary schools (49.48%) and 771 secondary schools (50.52%) in Shanghai (China), from a socio-ecological perspective, including individual aspects, social support, and physical environment. The results of this research are aligned with those obtained in the previous research analyzed in this review [36,37]. The authors of this study emphasize that schools should provide more facilities for playing TT in a manner similar to what has been indicated in other studies [48,52]. In addition, it is suggested that communities play a fundamental role in this regard, so they should also provide more sports infrastructures to support adolescent exercise. The idea of Xiao et al. (2020) [38] is to use the sport of TT to increase its practice inside and outside the school environment, in order to reduce the tendency of students not to do physical exercise, trying to promote the practice of physical activity through TT [44,46,53].

Something recurrent so far, and already pointed out by Hoffmann et al. (2018) [37], is the fact that researchers denounce an insufficient number of facilities for the practice of TT in schools. In addition, in line with the arguments of Greenwood et al. (2001) [36], Xiao et al. (2020) [38] indicate that PE teachers should respect the ideas of adolescents, that is, listen to their opinions and encourage them to participate in TT training, inside or outside the school environment. However, the existence of scarce teacher training on this sport is mentioned, and it is also stated that there are very few TT courses in schools [54], so improving the scientific and methodological training of teachers on this sport may be essential to be able to develop it properly, as pointed out by various authors [19,48,50]. In short, Xiao et al. (2020) [38] state that schools should improve PE and offer more opportunities for spontaneous games [55], such as the practice of TT. This statement is expressed by different studies [6,56], indicating that priority should be given, as Dollman (2018) [57] indicates, to effective interventions and policies aimed at improving the enjoyment of physical exercise, increasing the availability of sports facilities and increasing the level of social support, in order to facilitate and promote the creation of healthy habits of physical activity among young people [16,19,44,58]. The authors highlight that policymakers should offer a wide range of outdoor physical activities, especially TT, from which students can choose, which may be critical in establishing intrinsic motivation and persistence in present and future physical activity [59] (Brug et al., 2017). These three studies analyzed [36,37,38], show that, in addition to student preferences, teaching decisions significantly shape the content of PE, highlighting the critical role of professional preparation [53]. Integrating both approaches, teacher training and preference surveys, could result in a more robust and engaging curriculum, especially for racquet sports such as TT [44], suggesting that the quality of teaching depends on both the competence of the educator and their alignment with students’ interests [53,58].

A total of four didactic experiences in which the sport of TT is examined as educational content were included in this review. Arndt (1987) [39] proposes a novel TT practiced at a round table. This innovation solves some of the problems presented by traditional TT in the school environment, such as space (increasing the number of players per unit of area), a reduced number of players per table (incorporating more players who participate simultaneously), the dimensions of the playing surface (adapting the height and length of the game table by reducing it), and finally, to avoid possible injuries derived from its practice (eliminating corners by transforming the usual gaming table for a round one). Arndt (1987) [39], in addition to incorporating this novel educational adaptation, also adds an affective dimension to its game dynamics, promoting the use of TT, a series of values such as peace and international understanding, stimulating the overcoming of cultural barriers and fostering social integration in a similar way to other studies [14,60]. At the educational level, the use of a round TT is fascinating for young people and interesting for teachers because of the benefits it entails, by increasing the economy of the teaching effort. On the other hand, the round table is also very practical by reducing the necessary investment costs, basically by increasing the number of players per table, making TT accessible to a greater number of students. In short, the round table provides educational advantages, both organizational and sociological, by integrating the individual in group play activities where the acquisition of values is integrated and encouraged.

Schwager et al. (2012) highlight that PE teachers often face challenges that classroom teachers do not experience, and these challenges can affect their ability to work towards the learning objectives they have for their students, such as sometimes not having a gym to develop PE classes [42]. Considering this problem, it is of interest to seek new opportunities to work with students in PE classes [60]. One of these alternatives is table top tennis, a sports activity adapted from TT, similar to that proposed by other authors [6], which allows it to be taught in the classroom or in the dining room, and can provide opportunities for students to work towards the achievement of some of the affective objectives related to sportsmanship and personal responsibility. Schwager et al. (2012) [42] describe table top tennis as an innovative educational didactic experience that requires very little material and space to be used in PE classes in primary education [42]. The only equipment needed to play this adapted TT is tables or class desks for practice, foam or nerf balls, and a rope or adhesive tape for the nets. The goal of TT is to hit the ball over the net inwards (on the table or desk) so that the opponent cannot return it. In this game, students can use the palm (right) or the back (backhand) of either hand as a racket, so it is not even necessary to have an implement to hit the ball.

Table top tennis can provide opportunities for students to work towards achieving some of the affective objectives related to the development of social skills [14]. In this sense, Schwager et al. (2012) [42] propose table top tennis as a teaching strategy in PE classes that offers the opportunity to intentionally promote the development of self-confidence, teamwork, responsibility, respect, and sportsmanship of students. Inclusion is another essential pillar in PE [42]. Inclusion is another essential pillar in PE. Healy (2013) [40] addresses this issue by exploring the adaptation of equipment and materials to teach object control skills to students with disabilities through the activity called balloon TT, an adapted TT that consists of hitting a balloon with a racket, allowing the process of teaching fundamental motor skills to become a basic framework for the development of future movements [17,61], or of more complex sports and game skills [62]. The adaptations proposed by Healy (2013) [40] through the inclusion of balloon TT are based on the five principles of adaptation, i.e., size, sound, support, surface, and speed, considered as a practical framework for personalizing learning. For example, adjusting the size of a racket or adding sound to a ball can facilitate the participation of students with visual or motor limitations, transforming barriers into opportunities, while these modifications also stimulate creativity and interdisciplinary thinking by requiring collaboration with specialists in other areas. Compared to Greenwood et al. (2001) [36], who advocate for adapting the curriculum to preferences, Healy (2013) [40] extends this idea to the physical personalization of the learning environment, highlighting that adaptations not only improve motor performance [7,14], but also enrich the educational experience [15]. However, Healy (2013) [40] identifies a significant gap, such as the lack of studies on the availability and impact of adapted equipment in schools, suggesting practical challenges such as costs, teacher training, and access to resources [48]. Together, these approaches underscore the importance of flexibility in PE, whether through activity selection or tool modification, to cater to a diverse student population, connecting content personalization with environment accessibility to maximize engagement and motor development [9,13,63].

Finally, the last experience analyzed in this review is TT triathlon [41]. This activity has a multidisciplinary character where TT is integrated into a so-called triathlon that combines psychomotor skills (TT), cognitive skills (learning English), and affective skills (social responsibility), under the pedagogical model of sports education of Siedentop, Hastie, and van der Mars (2011) [64] in PE. In this experience, the social responsibility of the students is promoted through the assignment of individual roles within a team where English language grammar and vocabulary are practiced, also participating in online games that aim at a social and solidarity action such as donating rice to the United Nations World Food Program. This didactic experience highlights how PE, through the use of TT, can even transcend its traditional domain to collaborate with classroom teachers and promote global citizenship. These approaches converge on the idea that TT, adaptable in form and purpose, can meet multiple objectives from cultural integration to holistic development, depending on the pedagogical context [60]. The convergence of these studies suggests a comprehensive framework for modern PE based on personalization, inclusion, and innovation, where student preferences [36] should guide the selection of activities, balancing with teaching capacities [37] and inclusion needs [40], while TT, in its various forms, whether in a round table [39], traditional [42], or conceived as a triathlon [41], illustrates how a single activity can be adapted to meet multiple objectives, supported by socio-ecological factors such as interest and environment [38]. A common challenge is the gap between theory and practice, with a lack of resources [40,48], variability in teacher training [37,48,50,53], and structural limitations [38,48], requiring creative solutions, such as incorporating low-cost adaptations [37,42,60] or interdisciplinary collaboration [41]. Future research could explore how these strategies are applied in diverse contexts, assessing their long-term impact on participation in PE and sports activities that promote students’ well-being and future health.

Considering the studies included in this systematic review, it is necessary to mention the limitations derived from the type of publications that focus on the TT curriculum in primary schools. Firstly, the educational curriculum in PE is regulated in each country in a different way, allowing in some cases to contemplate modifications and adaptations, while in other geographical areas it can be totally closed, which limits the incorporation of TT, so the results obtained may not be generalizable. On the other hand, in the different studies reviewed, no mention has been made of the methodological quality of the studies, the type of study, the representativeness of the sample, or the type of population analyzed, being very varied and heterogeneous, so it is very difficult to standardize the results obtained. Finally, this review has only considered studies written in English and Spanish, an aspect that may considerably reduce the number of studies potentially to be considered in this review. However, the evidence found in the different databases analyzed allows us to have an initial vision of interest to know and answer the research question posed about the use of TT as content in PE classes in primary education. From a PE perspective, incorporating table tennis (TT) into school curricula may serve as an effective strategy to support children in meeting the World Health Organization’s recommendation of at least six minutes of daily moderate-to-vigorous physical activity. Given its affordability and minimal space requirements, TT presents a practical solution for schools operating with limited facilities or budgets. Future investigations should assess the sustained impact of TT on physical and developmental health indicators, such as body mass index, motor coordination, and psychological well-being, within school-aged populations.

## 5. Conclusions

The incorporation of TT in primary schools remains limited, and there is a notable lack of research focusing on its implementation. Although the PE curriculum has evolved to include new sports that can enhance students’ motivation, data from this systematic review indicates that TT is not widely featured in PE programs despite its numerous benefits. Teachers find the inclusion of TT in PE classes to be engaging, and students express a strong interest in practicing the sport.

TT can be successfully adapted to various contexts, as evidenced by the articles included in this systematic review. These include round TT, table top tennis, TT with balloons, TT triathlon, and traditional TT, all of which have effectively utilized the sport in primary school settings.

However, several challenges hinder the use of TT in primary PE. These challenges include a shortage of equipment, inadequate facilities, and a lack of teacher training to effectively teach the sport. Nevertheless, the didactic experiences analyzed in this review demonstrate that TT is a versatile and engaging option for inclusion in school PE programs.

Incorporating TT allows for an innovative, comprehensive, and inclusive approach to physical education. It can be tailored to fit various pedagogical situations and can support the development of important social values such as responsibility, inclusion, and solidarity, and can also enhance physical and psychological skills (coordination and motor skills) while improving overall physical fitness. 

## Figures and Tables

**Figure 1 sports-13-00251-f001:**
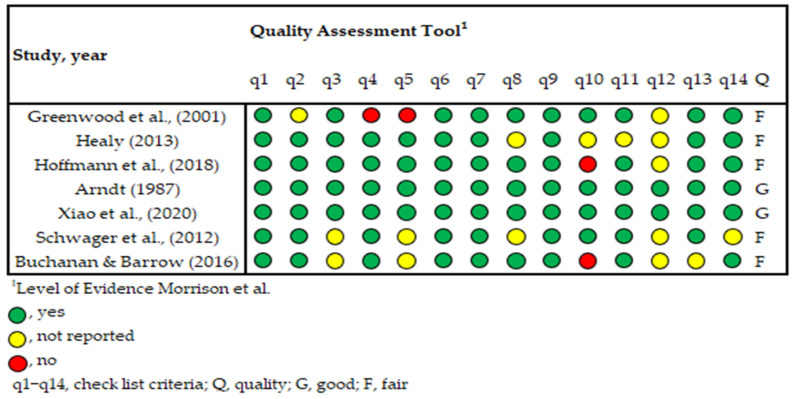
Methodological quality appraisal of studies included [36,37,38,39,40,41,42].

**Figure 2 sports-13-00251-f002:**
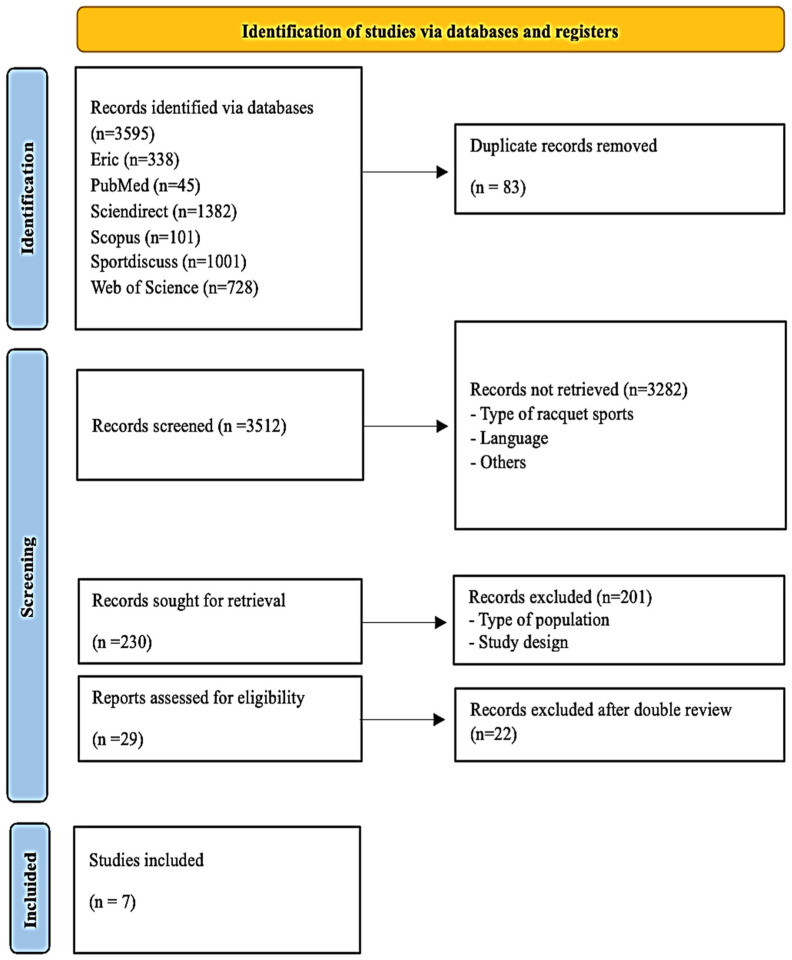
A flow chart describing the selection process of the included studies.

**Table 1 sports-13-00251-t001:** Inclusion and exclusion criteria for systematic review.

No.	Criteria	Inclusion	Exclusion
1	Population	Primary school healthy students	Other educational stages (Secondary, adults, etc.) and students with any medical condition or disability
2	Study Design	Descriptive, intervention studying exclusively tennis table	Studies analyzing other racket sports
3	Outcomes	Academic variables related to the content of tennis table	Variables not directly related to the learning process
4	Others	English or Spanish language	Other languages

**Table 2 sports-13-00251-t002:** Selected table tennis articles.

Author and Year	Study Population	Objective	Study Design	Intervention	Results	Conclusions
Arndt (1987) [39]	N = UN * S = UN * A = UN * TP = Physical educator teachers C = New Zealand	The aim of the study is to demonstrate that the new TT system, based on a circular table, offers special advantages and motivation for its practice. The study presents the main characteristics of the game and its implications on sociological, psychological, philosophical, and cultural aspects.	DEX	Design a round TT for youth with basic rules of play that solve the problems of conventional play.	It increases the number of players simultaneously. The nets are arranged radially and can be adjusted to any point on the perimeter of the table, modifying the size of the playing areas depending on the skill level of the players. It can be practiced with a single ball or several simultaneously. This game has many organizational and psychological advantages, as well as a multitude of tactical and gameplay possibilities.	Round TT allows physical activity to be promoted by coexisting with technology, so it is quite recommendable for children and young people today. It represents a new valid form of sports and recreational practice, so authors recommend its introduction in educational centers worldwide.
Greenwood et al. (2001) [36]	N = 751 S = Boys and girls A = Middle school TP = Students C = United States	The study aimed to determine the physical activity preferences of boys and girls.	DST	Complete an “Activity Interest Inventory” over a two-day period during the spring semester.	Total activity preferences: TT 243 (32.4%) strong interest; 271 (36.1%) undecided; 237 (31.6%) little interest. Male activity preferences: TT 131 (33.8%) strong interest; 145 (37.4%) undecided; 112 (28.9%) little interest. Female activity preferences: TT 112 (30.9%) strong interest; 126 (34.7%) undecided; 125 (34.4%) little interest.	Students differ and coincide with their interests in the selection of specific sports activities. It would seem reasonable to allow students to have a say in the inclusion of sports activities in the PE curriculum. The PE teacher should be cautious in selecting activities that can be taught within the specific constraints of the school setting (facilities, space).
Schwager et al. (2012) [42]	N = UN * S = UN * A = Students in grades four and up TP = Physical educator teachers C = UN *	The study describes a game that can be played anywhere (classroom or cafeteria) and that allows students to understand the importance of practicing fair and responsible play.	DEX	The rules of the game of TT (serving, rallying, and scoring) are described, along with strategies and tactics with the idea of promoting responsibility and sportsmanship, through the inclusion of differentiated roles, responsibilities of the players, and multiple ways to achieve the success of the team. Each team selects a captain, a coach, and a statistician by signing a contract in which they agree to respect fair play and contribute to the team’s success.	Table top tennis is a game similar to TT, playable in a classroom or dining room with minimal equipment. It incorporates affordable skills for all students. The necessary materials are tables or desks, foam balls, and rope or adhesive tape. The goal is to hit the ball over the net onto the table so that the opponent cannot return it. The key question is: “What do you hit the ball with if you don’t use a racket?” Students can use the palm (right) or back (back) of either hand. Depending on the number of students and the number or size of tables, it can be played in teams, singles, or doubles.	PE teachers must overcome challenges to meet educational standards that classroom teachers do not experience. If there is no gym available, TT (with an educational format) is an activity that can be taught in the classroom or in another adapted location. It offers opportunities for students from fourth grade onwards to work on achieving some of the affective goals related to sportsmanship and personal responsibility.
Healy (2013) [40]	N = UN * S = UN * A = Infancy TP = Physical educator teachers C = UN *	The aim of this intervention is to provide likely measures to adapt TT to the PE context.	DEX	Adapt task constraints by adapting the materials used, especially for children with disabilities, to help them acquire fundamental motor skills.	TT is an activity that involves hitting. Your game can be adapted by tying a balloon to the net with a string that reaches the edge of the table. This new situation adapts the speed and size, making it easier to play. It also allows you to play it on the floor or on a table, allowing the child to kneel, sit, or stand.	By using adapted equipment (e.g., TT with a balloon), children can learn, practice, and develop their motor skills to the fullest. These material adaptations allow for the development of inclusive PE and the preparation of physical educators to work with students with disabilities.
Buchanan & Barrow (2016) [41]	N = UN * S = UN * A = Primary school TP = Physical educator teachers C = United States	To describe the integration of three learning domains: cognitive (English language [ELA]—grammar and vocabulary), affective (social responsibility), and psychomotor skills (TT using the pedagogical model of sports education).	DEX	The TT triathlon is a team competition where you work by roles. Students establish who will play each role. Students work together to create a name for the team. Each team will learn and compete in table tennis to earn points. Each encounter will consist of a TT match, a vocabulary and grammar challenge, and an online challenge in freerice.com. To win the match, the teams work to succeed in all aspects: TT, ELA, and social responsibility.	PE—Psychomotor Mastery. It is a game that can be implemented inexpensively by buying only rackets, balls, and nets. English Language and Literature—Cognitive Mastery. The conventions of standard English and the acquisition and use of vocabulary were chosen as appropriate for integration into the sports education unit. Social Responsibility—Affective Domain. The pedagogical model of sports education focuses on social responsibility, since students must work as a team and each one must responsibly play their specific roles. Social responsibility is being promoted both locally and globally.	Students and teachers benefit from the integration of content areas by showing how PE content integrates seamlessly with other knowledge areas, improving children’s overall learning. At the school level, content integration is a way to participate in a collaborative teaching model. Incorporating social responsibility helps young people focus on the needs of others, whether at the team level or on a global level. An integrated, well-implemented curriculum benefits everyone, but especially students.
Hoffmann et al. (2018) [37]	N = 498 S = 97 male and 401 females A = 44.58 ± 10.56 years Primary school TP = Physical educator teachers C = Germany	To investigate the parameters that influence the decision to teach specific sports, in particular RS, in primary PE. It is hypothesized that (1) racquet sports are taught in PE classes, and (2) specific parameters (e.g., work experience, gender, and teacher qualifications) influence the decision to teach racquet sports as educational content.	DST	A standardized questionnaire was designed and subdivided into five sections: general information about participation, the educational center, the faculty, the framework (location and materials), and PE (e.g., guidelines and personal preferences for the implementation of racquet sports). Different types of questions are included (single-choice, multiple-choice, and open-ended).	The teachers’ responses indicate that 72.89% (363) of the schools have internal curricula. Within these curricula, 67.97% (244) include RS. A total of 69.88% (348) of the teachers teach RS classes in primary PE, divided into different types: basic games (59.77%), badminton (54.6%), TT (20.59%), and tennis (14.66%).	Two out of three primary school teachers teach RS in their PE classes, particularly basic games and badminton. Gender had no impact on the decision to teach RS, The study identified four specific parameters that influence the implementation of the teaching of RS in primary schools: the school’s internal PE curriculum; work experience; being a specialist in PE; and personal affinity for practicing RS during free time. It is suggested that a change in study regulations, for example, by offering extra-occupational courses to teachers, may benefit the implementation of RS in primary schools.
Xiao et al. (2020) [38]	N = 1526 Primary students: 755 (49.48%) Secondary students: 771 (50.52%) S = UN * A = 9 to 16 years (12.31 ± 1.32 years) TP = Physical educator teachers C = China	To explore the factors affecting adolescent TT skills (ATTS) in Shanghai from a socio-ecological perspective, including individual factors, social support, and physical environment. The following hypotheses were raised: (1) individual factors (motivation, self-efficacy), social support (support from parents, friends, PE teacher), and physical environment (school and community) have an impact on the performance of the ATTS test; (2) among levels of social ecology, different levels have different degrees of impact on ATTS improvement; and (3) in addition to the direct effect, the influence of individual factors on ATTS test performance has a mediating effect.	DST	Participants completed a questionnaire based on social ecological theory after taking the ATTS test.	Individual factors played an important role in improving ATTS test scores; (2) social support was positively related to ATTS test score; and (3) physical environment was also significantly associated with ATTS test score. The results of the multiple linear regression and SEM analysis showed that, among the three levels, individual factors were the most important for the improvement of ATTS. Self-efficacy was identified as an important factor related to ATTS test score.	The factor that most influences ATTS is individual factors, followed by social support, and the one that influences the least is the physical environment. Therefore, fostering intrinsic interest is critical to facilitating teens’ continued activity. Second, friends should support each other, and parents should properly encourage teens regarding the practice of TT. Schools should provide more facilities for playing TT. PE teachers should respect the ideas of teenagers, listen to their opinions, and encourage them to participate in TT training.

UN * = Unregistered.; N = Number of Subjects; S = Sex; A = Age; TP = Type of Population; C = Country; DST = Descriptive Study; DEX = Didactic Experience; RS = Racket Sports; PE = Physical Education; TT = Table Tennis.

## Data Availability

The dataset can be made available upon request.

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
