# Peer review of "Table Tennis as a Tool for Physical Education and Health Promotion in Primary Schools: A Systematic Review"

_sports, 2025, doi:10.3390/sports13080251_

Round 1
Reviewer 1 Report
Comments and Suggestions for Authors
This article presents a systematic review of the implementation and use of table tennis in primary school physical education curricula. This is a highly relevant and under-researched area, especially given the broader pedagogical impact of racket sports in early childhood education. The abstract summarises the findings well but should include information on the specific sample size and number of people included. The introduction, although the purpose is clear, would benefit from a stronger positioning in global trends in physical education curricula.The manuscript does not provide sufficient detail regarding the rationale for conducting a new systematic review. It would be helpful if the authors explained more clearly why an updated or separate review is warranted at this time, especially in light of previous evidence syntheses on similar topics. Although reference is made to PRISMA guidelines, key elements such as inclusion/exclusion criteria, coding reliability and assessment of risk of bias are not sufficiently described. The results are informative and well categorised, but no attempt is made to quantitatively synthesise or meta-synthesis. The discussion is appropriate but requires better linkage to recent pedagogical research. Cultural or policy-based variability across countries is not reflected. Limitations are poorly highlighted. Linguistically clear and grammatically correct. Minor improvements are recommended. Figures/Tables PRISMA flow chart is included (good), but synthesis tables could be more detailed.
Comments on the Quality of English LanguageAccept after minor revisions
Author Response
Response to Reviewer 1 Comments
Dear reviewer 1,
Thank you for giving us the opportunity to submit a revised draft of our manuscript titled "Table Tennis as a Tool for Physical Education and Health Promotion in Primary Schools: A Systematic Review” to the issue “Promotion of Healthy Habits, Inclusive Sport and Educational Improvement”.
We appreciate the time and effort that you have dedicated to providing your valuable feedback on our manuscript. Consequently, we have been able to incorporate changes to reflect most of the comments provided by you. We have highlighted the changes within the manuscript.
Here is a point-by-point response to your main notes and concerns. All these changes have been added to the main document and we have highlighted the corrections in green colour for you to find them easily. Additionally, some comments have been inserted. We hope you find it helpful.
Comment:
The abstract summarises the findings well but should include information on the specific sample size and number of people included.
Response:
We agree and we added the information requested in the abstract. We have added the total population included, however the studies included do not discriminate the characteristics of the population included (sex, age, etc.)
Comment:
The introduction, although the purpose is clear, would benefit from a stronger positioning in global trends in physical education curricula.
Response:
We fully agree we have added two lines (38-40) explaining new trends like meaningful physical education and its relationship with students´ motivation.
Comment:
The manuscript does not provide sufficient detail regarding the rationale for conducting a new systematic review. It would be helpful if the authors explained more clearly why an updated or separate review is warranted at this time, especially in light of previous evidence syntheses on similar topics. Although reference is made to PRISMA guidelines, key elements such as inclusion/exclusion criteria, coding reliability and assessment of risk of bias are not sufficiently described.
Response:
This article presents a systematic review of the implementation and use of table tennis in primary school physical education curricula. This is a highly relevant and under-researched area, especially given the broader pedagogical impact of racket sports in early childhood education.
Inclusion and exclusion criteria are fully described in table 1 using the standardized PICO system. Risk of bias and quality assessment was done through and specific tool for education and didactic experiences instead of PEDRO or COCHRANE that are not designed or validated for these types of studies. Both items are fully described in independent sections (2.3 and 2.4).
To our knowledge, there is currently no systematic review in the existing literature that investigates the use of table tennis at the educational level. This study is the first of its kind, highlighting the need for a systematic review as an initial examination of the available scientific evidence in this area of pedagogy and sports. As a pioneering effort focusing on existing evidence, our objective was not to conduct a meta-analysis. In the introduction, we outline our reasons for pursuing this systematic review on the use of table tennis. Our interest is driven by a growing trend in recent years to analyze this sport and the scientific community's focus on its significant physical, psychological, and health benefits when practiced by children.
Comment:
The results are informative and well categorised, but no attempt is made to quantitatively synthesise or meta-synthesis. The discussion is appropriate but requires better linkage to recent pedagogical research. Cultural or policy-based variability across countries is not reflected. Limitations are poorly highlighted. Linguistically clear and grammatically correct. Minor improvements are recommended. Figures/Tables PRISMA flow chart is included (good), but synthesis tables could be more detailed.
Response:
We acknowledge your observation regarding the addition of some entire paragraphs from the articles. The reason for this is that the studies included in our work are mainly didactic experiences. Out of the seven studies, only three are descriptive, while the remaining four focus on didactic experiences that provide advice without offering descriptive results. The aim of this systematic review is to analyse globally the situation of TT in Primary Schools but no subgroup or additional objectives were established. As long as the number of studies focusing on TT keeps growing in the future we would be allowed to set metanalysis or more deep research.
Please let us know if any further revisions are required. Thank you again for your time and constructive feedback.
Kind regards,
Miguel Lecina
(on behalf of all co-authors)

Reviewer 2 Report
Comments and Suggestions for Authors
I find nothing inherently wrong with this paper. It is a form of data mining based on descriptive statistics. The authors did a search of particular variables concerning table tennis and elementary children, and teaching table tennis within the elementary school. So, in general the authors were looking for the incidence of research focused on elementary children and table tennis. The criteria was very broad and so they had numerous hits of which they did a sample of these articles and then reviewed the content of these articles. The authors reviewed seven articles from this general population of 3595 articles. And from those seven they tried to argue the need for table tennis as a fitness element in elementary school children. Even if one never read any of these articles, one could argue the premise of what the authors found. Table tennis could serve as an effective tool increasing children's physical activity levels, improve motor skills, and foster social-emotional development. The if is based on the quality of experience. Some basic errors that need to be addressed: The page numbers are not congruent and there are several of the same page numbers in the article; in other words, two page 5's and so forth. Also, the reference pages are not tidy or clean. Some journal articles are capped; and others are not. I am guessing the authors used a collector of some sort to capture the references. The problem with this sort of thing causes one to wonder if the authors are sloppy with their references, are they also sloppy with their data? As I said earlier, I don't think this article has majors errors, but then the authors are not creating new knowledge. The purpose statement of " a systematic review was to analyze the use of TT as educational content in the subject of PE in primary school". This is not a rocket science article. I am always suspect of this sort of research paper: it's really an analysis searching for a publication. I will give a thumbs up to this paper as long as authors clean the references and the page numbers.
Author Response
Response to Reviewer 2 Comments
Dear Reviewer 2,
Thank you for the opportunity to submit a revised draft of our manuscript titled “Table Tennis as a Tool for Physical Education and Health Promotion in Primary Schools: A Systematic Review” for the special issue “Promotion of Healthy Habits, Inclusive Sport and Educational Improvement.”
We sincerely appreciate the time and effort you have dedicated to providing valuable feedback on our manuscript. Based on your comments, we have made several revisions, which are highlighted in the manuscript. Additionally, we have included comments within the document for further clarification where necessary.
Below is a point-by-point response to your main observations and concerns. All corresponding changes have been incorporated into the revised version of the manuscript, and highlighted in green for ease of identification. We hope you find these adjustments helpful.
Comment:
Some basic errors that need to be addressed: The page numbers are not congruent, and there are several duplicate page numbers in the article (e.g., two pages labeled as “5”).
Response:
This issue has been resolved. The page numbering error was due to a formatting issue in the journal’s template, which we have now corrected.
Comment:
Also, the reference pages are not tidy or clean. Some journal article titles are capitalized inconsistently. I suspect the authors used a reference manager, but the inconsistency raises concerns about attention to detail. If references are handled carelessly, one might wonder if the same applies to the data. As stated earlier, I don’t think the article has major errors, but the authors are not generating new knowledge.
Response:
Thank you for pointing this out. The references have been carefully revised and reformatted to ensure consistency and adherence to the journal's style guide. The issue originated from our reference manager software, but we have now thoroughly reviewed each entry manually.
Comment:
The purpose statement—"a systematic review was to analyze the use of TT as educational content in the subject of PE in primary school"—suggests this is not a groundbreaking article. I’m often wary of papers that appear to be written solely for publication. That said, I am willing to support the paper provided the references and page numbers are cleaned up.
Response:
We acknowledge your perspective, and we respectfully offer a different view. In our opinion, this systematic review provides relevant and practical insights for both educators and students in the field of physical education.
As a secondary PE teacher with over ten years of experience, I have personally observed a growing disengagement among students toward traditional physical education. The rise of digital entertainment—such as screens, video games, and social media—has dramatically reshaped students’ free time and motivations. This evolving context challenges educators to adapt and seek more engaging, dynamic approaches to PE.
Table tennis is one such activity, offering high engagement potential. However, there is a noticeable lack of empirical evidence and pedagogical guidance on how to integrate it effectively into the primary school PE curriculum. This systematic review seeks to fill that gap—not simply to add another paper to the literature, but to provide educators with applicable, evidence-based recommendations that support innovation and improve student motivation.
This article aims to provide valuable insights into a topic of mutual interest for both table tennis and physical education, rather than merely adding another publication to the existing body of work. As a table tennis coach, a physical education professor, and a researcher at the university, my goal is to introduce innovative sports content that can enhance motor skills and promote physical activity. This, in turn, contributes to better health outcomes by offering more engaging and motivating activities that encourage students to be active from an early age, all from a scientific perspective. We hope to clarify that the primary intention of this article is not to seek publication for its own sake. Instead, we aim to initiate a methodologically sound approach to developing deeper research that can lead to a greater understanding of the benefits of practicing table tennis in educational settings.
Please let us know if any further revisions are required. Thank you again for your time and constructive feedback.
Kind regards,
Miguel Lecina
(on behalf of all co-authors)
Reviewer 3 Report
Comments and Suggestions for Authors
- The authors conducted interesting research, methodologically correct (PRISMA; PICO; two independent reviewers). In the Discussion, extensive abstracts of the selected articles.
- Basic question for authors: please indicate in the Discussion where your contribution, your analysis, your critical analysis, and your thoughts are. I appreciate your work; however, "Systematic review" is not just about selecting articles and rewriting abstracts. I kindly ask for an explanation.
- Conclusions: The first sentence should not be here, because it was not the aim of your research „One of the main findings of this systematic…”. Conclusions should be specific, and statements like “different social, physical and psychological skills” should be avoided. This (“different”) should be specified. The following sentences, “The use of TT in its different formats produces benefits in students in the psychomotor, cognitive and affective domains”, and again, “different” should be specified. Was there only “different” in the articles selected by the authors? In general, conclusions need to be refined.
- Incorrect number spelling, e.g. "3595", should be "3,595".
Author Response
Response to Reviewer 3 Comments
Dear reviewer 3,
Thank you for giving us the opportunity to submit a revised draft of our manuscript titled "Table Tennis as a Tool for Physical Education and Health Promotion in Primary Schools: A Systematic Review” to the issue “Promotion of Healthy Habits, Inclusive Sport and Educational Improvement”.
We appreciate the time and effort that you have dedicated to providing your valuable feedback on our manuscript. Consequently, we have been able to incorporate changes to reflect most of the comments provided by you. We have highlighted the changes within the manuscript.
Here is a point-by-point response to your main notes and concerns. All these changes have been added to the main document and we have highlighted the corrections in red colour for you to find them easily. Additionally, some comments have been inserted. We hope you find it helpful.
Comment:
- The authors conducted interesting research, methodologically correct (PRISMA; PICO; two independent reviewers). In the Discussion, extensive abstracts of the selected articles.
Response:
We acknowledge your observation regarding the addition of some entire paragraphs from the articles. The reason for this is that the studies included in our work vary in type. Out of the seven studies, only three are descriptive, while the remaining four focus on didactic experiences that provide advice without offering descriptive results.
Comment:
- Basic question for authors: please indicate in the Discussion where your contribution, your analysis, your critical analysis, and your thoughts are. I appreciate your work; however, "Systematic review" is not just about selecting articles and rewriting abstracts. I kindly ask for an explanation.
Response:
We offer our point of view regarding the tennis table in lines (258-266), our contribution (lines 198-206) applicability TT (lines 227-232); your contribution, our analysis (lines from 399 - 406). Please read carefully the aforementioned lines where all your questions are answered. We acknowledge your observation regarding the addition of some paragraphs. The reason for this is that the studies included in our work vary in type. Out of the seven studies, only three are descriptive, while the remaining four focus on didactic experiences that provide advice without offering descriptive results.
Comment:
- Conclusions: The first sentence should not be here, because it was not the aim of your research „One of the main findings of this systematic…”. Conclusions should be specific, and statements like “different social, physical and psychological skills” should be avoided. This (“different”) should be specified. The following sentences, “The use of TT in its different formats produces benefits in students in the psychomotor, cognitive and affective domains”, and again, “different” should be specified. Was there only “different” in the articles selected by the authors? In general, conclusions need to be refined.
Response:
We have rewritten the first sentences of the conclusion paragraph following your advice. We have removed the duplicate adjective dfferent
The different sports related to TT are detailed in parentheses and the social, physical and psychological skills indicated in the studies analysed are also incorporated.
Comment:
- Incorrect number spelling, e.g. "3595", should be "3,595".
Response:
The numbers have been revised to adapt to the English spelling.
Please let us know if any further revisions are required. Thank you again for your time and constructive feedback.
Kind regards,
Miguel Lecina
(on behalf of all co-authors)
